# Phage predation accelerates the spread of plasmid-encoded antibiotic resistance

Chujin Ruan [1,2], Josep Ramoneda [3,4], Anton Kan [5], Timothy J. Rudge[6], Gang Wang [1,7] ✉ & David R. Johnson [2,8] ✉

Phage predation is generally assumed to reduce microbial proliferation while not contributing to the spread of antibiotic resistance. However, this assumption does not consider the effect of phage predation on the spatial organization of different microbial populations. Here, we show that phage predation can increase the spread of plasmid-encoded antibiotic resistance during surface-associated microbial growth by reshaping spatial organization. Using two strains of the bacterium *Escherichia coli*, we demonstrate that phage predation slows the spatial segregation of the strains during growth. This increases the number of cell-cell contacts and the extent of conjugation-mediated plasmid transfer between them. The underlying mechanism is that phage predation shifts the location of fastest growth from the biomass periphery to the interior where cells are densely packed and aligned closer to parallel with each other. This creates straighter interfaces between the strains that are less likely to merge together during growth, consequently slowing the spatial segregation of the strains and enhancing plasmid transfer between them. Our results have implications for the design and application of phage therapy and reveal a mechanism for how microbial functions that are deleterious to human and environmental health can proliferate in the absence of positive selection.

Phage are integral components of microbial ecosystems that can direct ecosystem functioning and dynamics with consequences for human health, biotechnology, and elemental cycling[1–5]. The enormous influence of phage stems from them being the most abundant biological entity on Earth while also being effective predators[6–8]. They have highly selective host ranges, which can cause specific changes to microbial abundances, diversity, and interactions that can modify ecosystem functioning and stability[9–11]. They also impose strong selection pressures on their hosts that can drive ecosystem dynamics over ecological and evolutionary timescales[12–14].

The effectiveness of phage predation depends on whether the host is in a surface-associated state (e.g., biofilms, colonies, aggregates, etc.)[15,16]. This is typical as a large proportion of microbial life is associated with surfaces[17–19]. Surface association can increase phage predation by increasing local phage concentrations and the duration of physical contacts with host cells[20]. However, surface association can also repress phage predation by causing changes to host physiology and local environmental conditions[21,22]. This includes forming an extracellular matrix that can slow phage transport to host cells[22,23], creating regions of low metabolic activity that are less susceptible to

[1]College of Land Science and Technology, China Agricultural University, Beijing, China. [2]Department of Environmental Microbiology, Swiss Federal Institute of Aquatic Science and Technology (Eawag), Dübendorf, Switzerland. [3]Spanish Research Council (CSIC), Center for Advanced Studies of Blanes (CEAB), Blanes, Spain. [4]Cooperative Institute for Research in Environmental Sciences, University of Colorado, Boulder, CO, USA. [5]Department of Materials, Swiss Federal Institute of Technology (ETH), Zürich, Switzerland. [6]Interdisciplinary Computing and Complex Biosystems (ICOS) Research Group, School of Computing, Newcastle University, Newcastle upon Tyne, UK. [7]National Black Soil & Agriculture Research, China Agricultural University, Beijing, China. [8]Institute of Ecology and Evolution, University of Bern, Bern, Switzerland. ✉e-mail: gangwang@cau.edu.cn; david.johnson@eawag.ch

phage predation[24–26], and inducing the secretion of molecules that inhibit phage[24,27]. Surface association also enables the process of microbial spatial self-organization, whereby different microbial populations arrange themselves across space as a consequence of their traits, local environmental conditions, and interactions with neighboring cells[28–30]. This can result in spatial patterns that physically protect or expose host cells to phage[15,21,22,31]. Phage predation can also feedback on spatial self-organization and, in turn, modify local environmental conditions and interactions[32,33]. Thus, there is a complex interplay between phage predation and spatial self-organization that can determine the dynamics and functioning of surface-associated microbial ecosystems.

Here, we hypothesize that phage predation can drive the spread of plasmid-encoded antibiotic resistance by modifying microbial spatial self-organization. More precisely, we hypothesize that phage predation increases the spatial intermixing of different microbial populations during surface-associated growth, consequently increasing the number of cell–cell contacts and promoting conjugation-mediated plasmid transfer between them. Our hypothesis is based on fundamental principles of surface-associated microbial growth. Only those cells located at the biomass periphery typically have access to

resources replenished from the environment, and only those cells therefore grow and contribute to new biomass (Fig. 1a; referred to as the active layer)[28,34,35]. Because their population sizes are small, they are subject to stochastic fluctuations that cause different microbial populations to spatially segregate along the biomass periphery (Fig. 1a), which is referred to as spatial demixing[28,35–37]. Briefly, the interfaces between microbial populations stochastically meander during surface-associated growth, which can cause neighboring interfaces to merge together (Fig. 1a). This reduces the number of cell–cell contacts between different microbial populations and the probability that plasmid transfer will occur between them[38,39]. However, phage have more ready access to, and are therefore more likely to predate on, cells located at the biomass periphery, which are the cells undergoing the most rapid spatial demixing (Fig. 1a). We therefore expect phage predation to slow spatial demixing, preserve more cell–cell contacts between different microbial populations, and promote plasmid transfer between them. Stated alternatively, we expect phage predation to hinder any one microbial population from dominating the biomass periphery and consequently increase spatial intermixing and plasmid transfer (Fig. 1a).

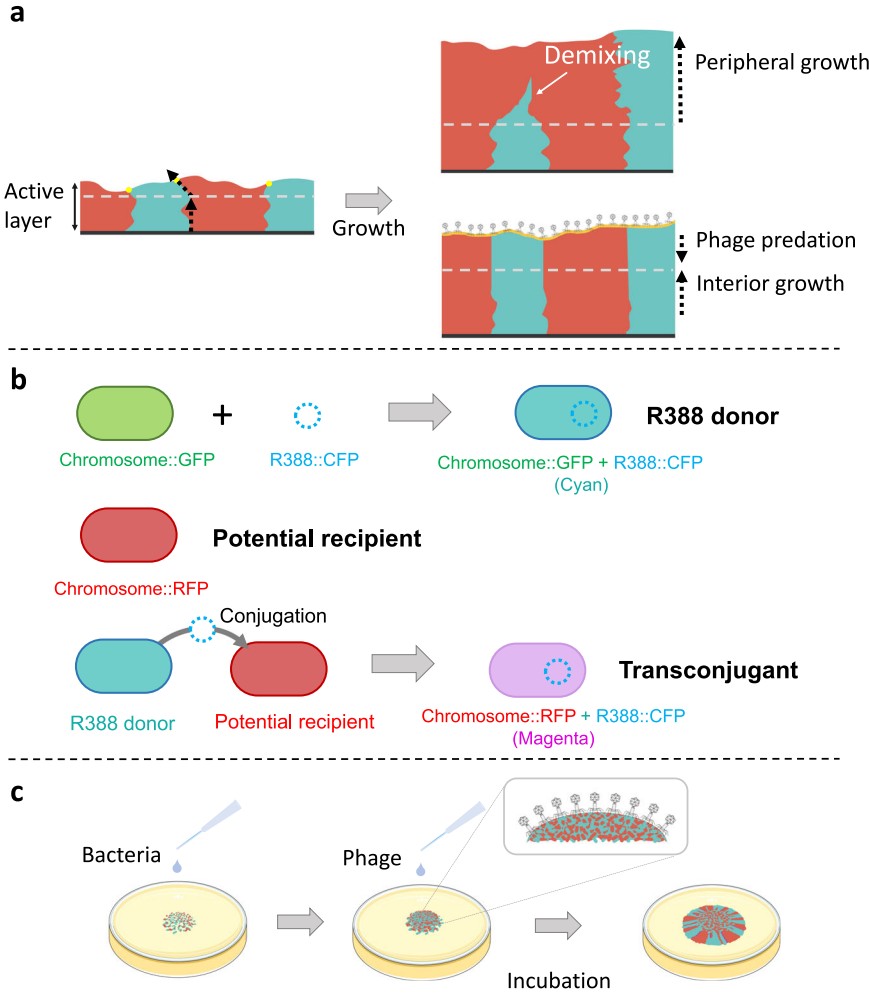

**Fig. 1 | Hypothesis and experimental system. a** We hypothesize that phage predation slows the spatial demixing of different microbial populations during surface-associated growth, consequently increasing plasmid transfer between them. Phage will preferentially predate on those cells located at the biomass periphery, which are the cells undergoing the most rapid spatial demixing, thus maintaining more spatial intermixing and enhancing plasmid transfer. **b** The biological components of our experimental system. The plasmid donor (designated as the R388 donor) expresses GFP from the chromosome and CFP from R388 and appears cyan. The

potential recipient expresses RFP from the chromosome. If R388 successfully transfers from the R388 donor to the potential recipient, the potential recipient will express RFP from the chromosome and CFP from R388 and appear magenta. **c** Our experimental approach is to grow the R388 donor and potential recipient together on a nutrient-rich surface in the presence or absence of the lytic phage T6 in anoxic conditions and quantify the effects of phage predation on the emergence of transconjugants.

To test our hypothesis, we performed surface-associated growth experiments with pairs of competing strains of the bacterium *Escherichia coli* in the presence or absence of phage. The strains can engage in the conjugation-mediated transfer of plasmid R388, which is self-transmissible and encodes for cyan fluorescent protein (CFP) and resistance to chloramphenicol (Fig. 1b)[40]. We refer to one strain as the R388 donor and the other as the potential recipient (Fig. 1b). We mix the strains together, grow them across nutrient-rich surfaces, infect them with the T6 lytic phage[41], and track the extent of R388 transfer using confocal laser-scanning microscopy (CLSM) (Fig. 1c). We complement our experiments with individual-based computational simulations that test how the peripheral killing caused by phage predation can reshape microbial spatial organization and increase plasmid transfer during surface-associated growth.

## Results

### Phage predation increases R388 spread during surface-associated microbial growth

We first quantified the effect of phage predation on the spread of R388 as the R388 donor and potential recipient grow together across a nutrient-amended agar surface. We find that phage predation increases the spread of R388 even in the absence of positive selection for R388 in the form of added chloramphenicol (Fig. 2a, b). This is supported by four lines of evidence. First, the number of spatially discrete transconjugant regions (i.e., sectors composed of cells expressing CFP and RFP and appearing magenta) is larger when phage are present (two-sample two-sided Welch test; $P = 5.2 \times 10^{-8}$, $n = 5$) (Fig. 2c). Second, transconjugants comprise a greater proportion of the total biomass when phage are present (two-sample two-sided Welch test; $P = 6.0 \times 10^{-8}$, $n = 5$) (Fig. 2d). Third, the total number of transconjugants is larger when phage are present (two-sample two-sided Welch test; $P = 7.6 \times 10^{-6}$, $n = 5$) (Fig. 2e). This is true even though phage predation reduces the total biomass size (two-sample two-sided Welch test; $P = 1.8 \times 10^{-6}$, $n = 5$) (Fig. 2f). Finally, our outcomes remain valid when we calculate the number of transconjugants at a fixed radial distance across all of our samples (Supplementary Fig. 1). Overall,

nearly all of the potential recipients receive R388 when phage are present (Fig. 2a), despite the fact that transconjugants grow slower than their R388-free counterparts (R388 reduces the growth rate by ~5%)[42]. In contrast, only those potential recipients lying at the interfaces with the R388 donor typically receive R388 when phage are absent (Fig. 2b). Thus, when phage are present, the production of transconjugants exceeds their removal via phage predation and outcompetition by R388-free counterparts. These outcomes remain valid when we reduce the initial relative abundance of the R388 donor by up to eightfold (Supplementary Fig. 2), demonstrating that only a few R388 donors are needed for the positive effect of phage predation on R388 transfer to manifest. They also remain valid in oxic conditions (Supplementary Fig. 3a, b) and when we reduce the nutrient concentration by 90% (Supplementary Fig. 3c, d).

### Natural transformation and transduction are not important R388 transfer mechanisms

We next tested whether conjugation-independent mechanisms of horizontal gene transfer (i.e., natural transformation and transduction) can explain our results, as *E. coli* can acquire plasmid-encoded genes via conjugation-independent mechanisms[43]. To test this, we prepared a phage-induced lysate of the R388 donor and applied it to the potential recipient during surface-associated growth. We concurrently applied heat-inactivated phage to test for natural transformation or viable phage to test for transduction. For both treatments, CFP is undetectable throughout the entire biomass (Supplementary Fig. 4a), which is expected if natural transformation and transduction are negligible. We then suspended the biomass that received the lysate and streaked it onto chloramphenicol-amended agar plates. We do not observe any growth regardless of whether we apply heat-inactivated or viable phage, which is again expected if natural transformation and transduction are negligible. Finally, we repeated the experiments with DNase I to degrade free DNA. We do not find statistically significant evidence that DNAse I affects the number of transconjugants (two-sample two-sided Welch test; $P = 0.38$, $n = 5$) (Supplementary Fig. 4b), which is expected if natural transformation is negligible. Taken

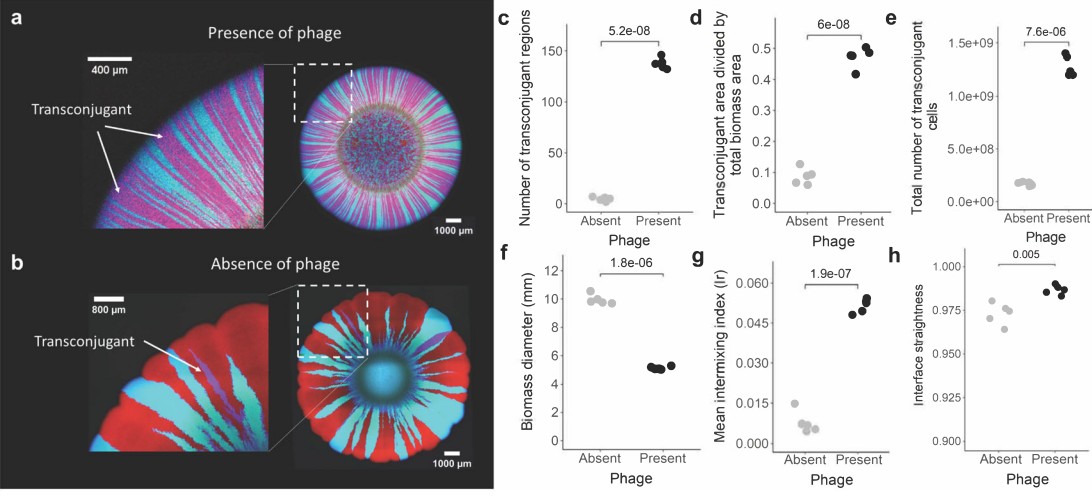

**Fig. 2 | Surface-associated growth experiments with the direct amendment of phage. a, b** Representative CLSM images ($n = 5$) of the R388 donor and potential recipient after ten days of growth in anoxic conditions in the (**a**) presence or (**b**) absence of phage, where we added the phage directly to the growing biomass. The R388 donor expresses GFP and CFP and appears cyan, the potential recipient expresses RFP and appears red, and transconjugants express RFP and CFP and appear magenta. The images on the left are magnifications of the regions in the white dashed boxes in the images on the right. **c** The number of discrete transconjugant regions (magenta sectors). **d** The total transconjugant area divided by

the total biomass area. **e** The total number of transconjugant cells quantified by colony counting on chloramphenicol-amended agar plates. **f** The biomass diameter. **g** The intermixing index quantified across a 30 μm thick band positioned at the biomass periphery. **h** The mean interface straightness. **c**–**h** Each data point is a measurement for an independent experimental replicate ($n = 5$) and the $P$ values are for two-sample two-sided Welch tests. Gray symbols are in the absence of phage and black symbols are in the presence of phage. Source data are provided as a Source Data file.

together, our data establish that phage predation does not promote the transformation or transduction of R388 or its associated genes, and that conjugation-independent mechanisms therefore cannot explain our results.

## Phage predation increases R388 transfer by slowing spatial demixing

Our hypothesis posits that phage predation slows the spatial demixing of different microbial populations along the biomass periphery (Fig. 1a), which in turn decreases the number of cell–cell contacts and the extent of R388 transfer between them. To test this, we quantified the effect of phage on the magnitude of spatial intermixing between the R388 donor and potential recipient. We use an intermixing index that quantifies the number of transitions between two strains (colors) adjusted for the anticipated number of transitions for a random spatial arrangement of the two strains. We first position a circle with its center located at the centroid of the biomass. We then calculate the intermixing index along the circumference of the circle as $I_r = (2N_r)/\pi r$, where r is the radius of the circle. This allows us to quantify the intermixing index as a function of the radial extent of growth. We find that spatial intermixing is significantly higher when phage are present (two-sample two-sided Welch test; $P = 1.9 \times 10^{-7}$, $n = 5$) (Fig. 2g), which provides evidence that phage predation does indeed slow spatial demixing. This effect remains valid when R388 is absent from the experiments (Supplementary Fig. 5), demonstrating that it is an R388-independent effect.

To provide further evidence that phage predation slows spatial demixing, we performed an experiment where we inoculate the phage at a distance of 1 mm from where we inoculate the mixture of the R388 donor and potential recipient on the nutrient-amended agar surface. The phage must therefore diffuse across the surface to predate on their host. We expect that the side of the biomass facing towards the phage inoculation area is exposed to more phage, and thus has higher spatial intermixing and more extensive R388 transfer, than the side facing away. Indeed, we observe higher spatial intermixing on the side facing toward the phage inoculation area (two-sample two-sided Welch test; $P = 0.019$, $n = 3$) (Fig. 3a, c). We also observed significantly more transconjugant regions on the side facing towards the phage inoculation area (two-sample two-sided Welch test; $P = 7.3 \times 10^{-5}$, $n = 3$) (Fig. 3a, d), which is expected if the extent of spatial intermixing determines the extent of R388 transfer.

Finally, we performed a third experiment where we reduce the dosage of phage applied to the biomass of the R388 donor and potential recipient (Supplementary Fig. 6). We expect that at a sufficiently low phage dosage, phage predation will be patchy across the biomass. This should generate correlated variance in the magnitude of spatial intermixing and transconjugant proliferation along the biomass periphery, where local regions with higher phage predation have higher spatial intermixing and more transconjugants. This is indeed what we observe (Supplementary Fig. 6).

## Phage predation slows spatial demixing by reshaping spatial organization

When analyzing the surface-associated growth experiments, it is evident that the interfaces between the R388 donor and potential recipient are straighter when phage are present (two-sample two-sided Welch test; $P = 0.005$, $n = 5$) (Fig. 2h). This is particularly evident when we inoculate the phage at a distal location, where the interfaces become significantly straighter immediately after contact with phage (Fig. 3b) and are straighter on the side of the biomass facing towards the phage inoculation area (two-sample two-sided Welch test; $P = 0.0008$, $n = 5$) (Fig. 3e). Straighter interfaces are less likely to merge together during surface-associated growth[28], thus slowing the spatial demixing of different microbial populations and maintaining more cell–cell contacts and promoting plasmid transfer between them.

Why does phage predation cause the formation of straighter interfaces? We hypothesize that phage predation shifts the location of fastest growth from the biomass periphery to the interior. In the absence of phage, cells at the biomass periphery grow the fastest due to their preferential access to resources replenished from the environment (Fig. 4a). In the presence of phage, cells at the biomass periphery are preferentially predated on, thus shifting the location of fastest growth to the interior (Fig. 4a). Importantly, the biomass periphery has lower cell packing densities and cells are aligned further from parallel (Fig. 4b), and we therefore expect peripheral growth to result in more meandering interfaces. In contrast, the interior has higher cell packing densities and cells are aligned closer to parallel (Fig. 4b), and we therefore expect interior growth to result in straighter interfaces.

To test this, we modified and employed an individual-based computational model where we assume that phage predation results in peripheral killing of the bacterial biomass. We implemented this approach because a prior study that simulated individual phage particles found that phage predation results in peripheral killing, where the depth of killing is related to population sizes and phage properties[31]. Using this approach, we find that peripheral killing accurately reproduces the effects that we observe in our experiments. When peripheral killing is absent in our simulations, cells at the biomass periphery grow the fastest. These cells are aligned further from parallel and form meandering interfaces that frequently merge together during surface-associated growth, resulting in rapid spatial demixing (Fig. 4c and Supplementary Video 1). In contrast, when peripheral killing is present in our simulations, cells at the biomass periphery are continuously removed, which causes cells in the interior to grow the fastest (Fig. 4d and Supplementary Video 1). These cells are aligned closer to parallel, which results in straighter interfaces (two-sample two-sided Welch test; $P = 2.3 \times 10^{-4}$, $n = 5$) (Fig. 4f) and higher spatial intermixing (two-sample two-sided Welch test; $P = 2.2 \times 10^{-16}$, $n = 5$) (Fig. 4g). Indeed, we find that cells are aligned close to parallel during phage predation in our experiments (Supplementary Fig. 7).

To further test this mechanism, we performed additional simulations without peripheral killing but where we prevent the two layers of cells at the biomass periphery from growing; we therefore directly set the location of fastest growth to the interior (Fig. 4e). This results in interfaces with straightness comparable to those observed with phage (mean interface straightness for interior growth = 0.993, SD = 0.0009; mean interface straightness with phage = 0.995, SD = 0.0003) (Fig. 4f and Supplementary Video 1). The magnitude of spatial intermixing is also comparable to that observed with phage (mean spatial intermixing for interior growth = −0.47, SD = 0.04; mean spatial intermixing with phage = −0.42, SD = 0.03) (Fig. 4g). Taken together, our simulations demonstrate that the key ingredient for the formation of straighter interfaces is to shift the location of fastest growth from the biomass periphery to the interior where cells have higher packing densities and are aligned closer to parallel, which reduces the probability that neighboring interfaces will merge together during surface-associated growth and slows spatial demixing.

## Peripheral killing lowers the conjugation rate needed to compensate for plasmid cost

Because conjugation-mediated plasmid transfer requires direct contact between a plasmid donor and a potential recipient cell, we expect transconjugants to emerge along the interfaces between those microbial populations. To test this, we integrated a plasmid transfer module into our individual-based computational model that sets a defined probability of plasmid transfer when a plasmid donor and a potential recipient cell come into physical contact. When peripheral killing is absent in our simulations, transconjugants exclusively localize along the interfaces between plasmid donor and potential recipient regions and do not substantially proliferate (Fig. 5a and

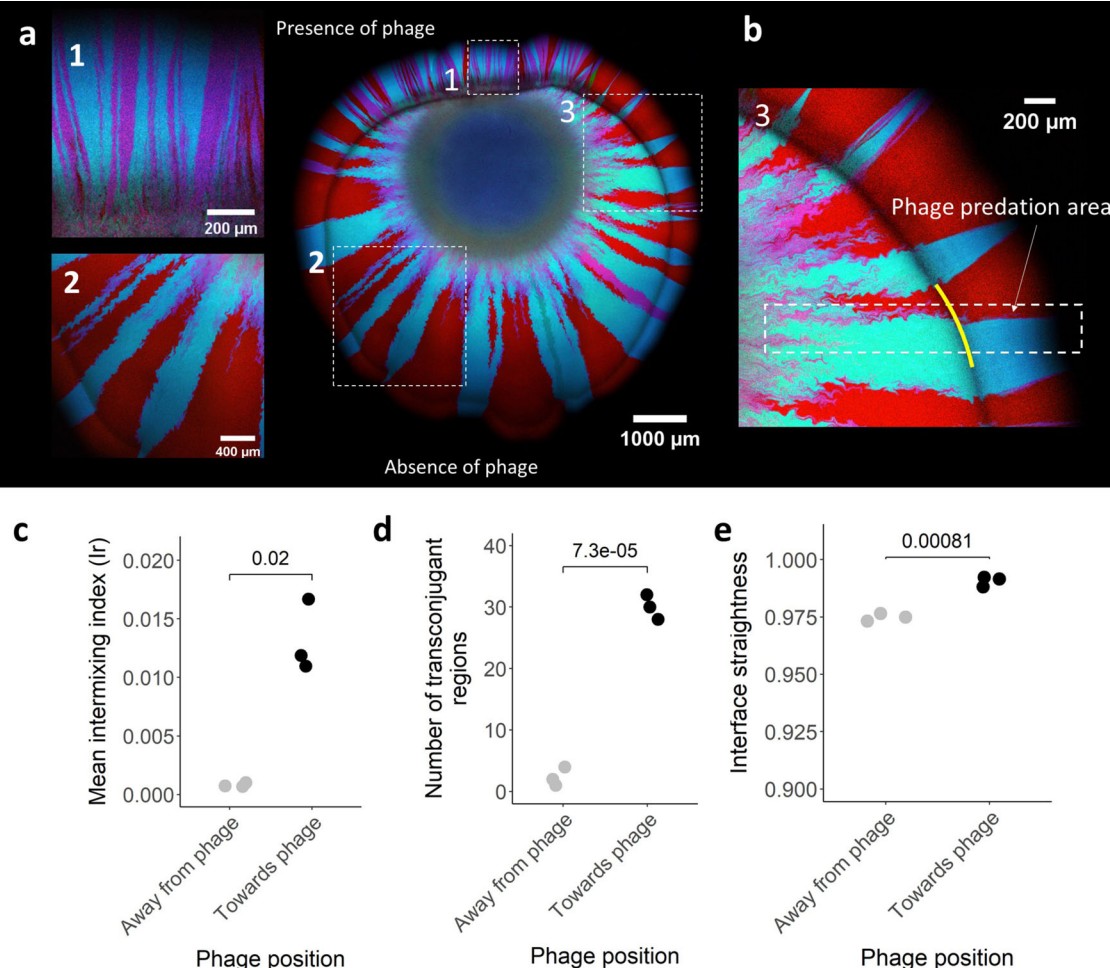

**Fig. 3 | Surface-associated growth experiments with the distal amendment of phage. a** Representative CLSM image ($n = 3$) of the R388 donor and potential recipient after ten days of growth in anoxic conditions where we added the phage at a distance of 1 mm in the positive $y$-direction from the centroid of the bacterial inoculation droplet. The R388 donor expresses GFP and CFP and appears cyan, the potential recipient expresses RFP and appears red, and transconjugants express RFP and CFP and appear magenta. The images on the left are magnifications of the regions in the white dashed boxes in the image on the right indicated by matching numbers. **b** Enlarged view of box 3 in (**a**) showing the change in interface straightness after phage encounter. The yellow line indicates the point in time when the phage made contact with the biomass. **c** The intermixing index at the biomass periphery. **d** The number of discrete transconjugant regions (magenta regions). **e** The mean interface straightness. **c**–**e** The quantities are for regions facing towards or away from the phage inoculation area. Each datapoint is a measurement an independent experimental replicate ($n = 3$), and the $P$ values are for two-sample two-sided Welch tests. Gray symbols are for the regions facing away from the phage inoculation area while black symbols are for the regions facing towards the phage inoculation area. Source data are provided as a Source Data file.

Supplementary Video 2), which is consistent with our experimental results (Fig. 2b). When peripheral killing is present in our simulations, transconjugants are not confined to the interfaces but instead proliferate and eventually displace potential recipients (Fig. 5b and Supplementary Video 3), which is again consistent with our experimental results (Fig. 2a).

Why do transconjugants proliferate when phage are present even though the plasmid reduces the growth rate? We hypothesize that the plasmid transfer probability and the large number of cell–cell contacts created by phage predation are sufficiently high to counteract the effects of out-competition by faster growing plasmid-free counterparts. To test this, we varied the plasmid transfer probability in our model between 0.0001 and 0.001. When peripheral killing is absent in our simulations, the number of transconjugants consistently increases as the plasmid transfer probability increases (Fig. 5c, d and Supplementary Fig. 8b). When peripheral killing is present in our simulations, there are always significantly more transconjugants regardless of the plasmid transfer probability (two-sample two-sided Welch tests; $P < 2.2 \times 10^{-16}$, $n = 5$) (Fig. 5c, d and Supplementary Fig. 8). Moreover, once the plasmid transfer probability exceeds 0.0003, nearly all of the

potential recipients receive the plasmid and the frequency of transconjugants approaches 0.5 within the entire biomass (Fig. 5c, d and Supplementary Fig. 8). Thus, when peripheral killing is present, a relatively low plasmid transfer probability can result in significant proliferation of transconjugants even though they grow slower than their plasmid-free counterparts.

## Discussion

Our findings identify a new consequence of phage-microbe interactions; phage predation can promote the conjugation-mediated transfer and proliferation of plasmids during surface-associated microbial growth by slowing the spatial demixing of different microbial populations. Unexpectedly, while phage predation decreases the total microbial biomass size, it can simultaneously increase the total number of transconjugants even though carrying the plasmid slows growth (Fig. 2). Our results therefore challenge the idea that predatory interactions inhibit plasmid transfer by reducing population sizes[44–46]. For example, phage predation can slow plasmid spread by creating an additional death rate[44], by specifically targeting cells that express plasmid-encoded traits[45], and by modifying selection pressures that

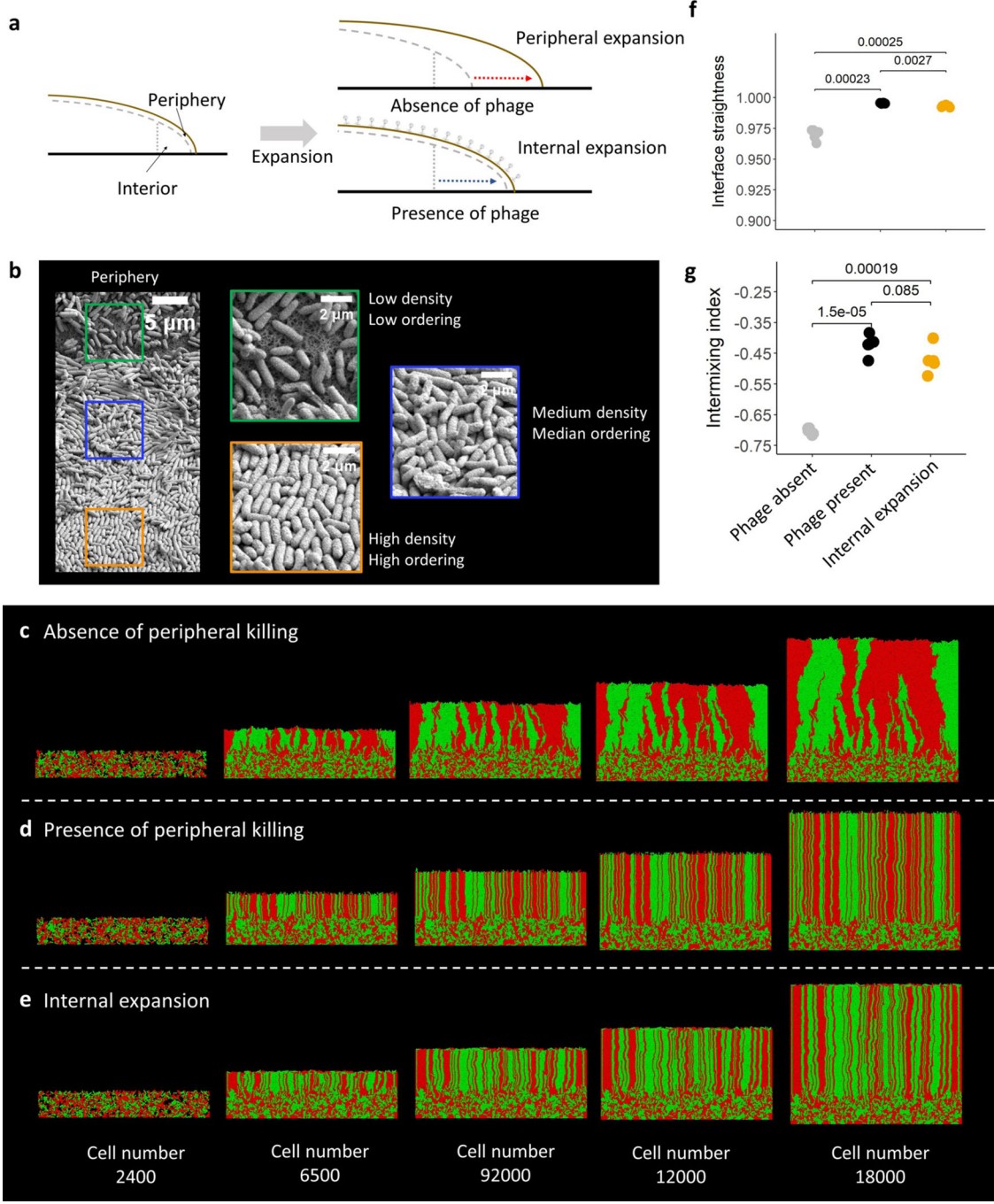

**Fig. 4 | Surface-associated growth simulations in the absence of plasmids.**
**a** Illustration of how phage predation shifts the location of fastest growth from the biomass periphery to the interior. When phage are absent, cells at the biomass periphery have preferential access to resources supplied from the environment, and they therefore grow the fastest. When phage are present, the cells at the periphery are removed by phage, thus causing cells located in the interior to grow the fastest. **b** Representative scanning electron microscopy image ($n = 5$) of the periphery of a colony growing in the absence of phage. The green box identifies cells located at the periphery, where they have low packing density and are aligned far from parallel. The blue box identifies cells located in the transition zone between the periphery and interior, where they have intermediate packing density and are aligned closer to parallel. The orange box identifies cells located in the interior, where they have high packing density and are aligned closest to parallel. **c**–**e** Representative simulations ($n = 5$) of two competing bacterial strains (green and red cells) that have the same growth rate. We simulated biomass growth until reaching a population size of 18,000 cells in the (**c**) absence or (**d**) presence of peripheral killing. **e** There is no peripheral killing but we set the growth rate of cells located in the outer two layers of the periphery to zero. This effectively sets the location of fastest growth to be in the interior. **f** The mean interface straightness. **g** The intermixing index. **f**, **g** Each datapoint is a measurement an independent simulation ($n = 5$) and the $P$ values are for two-sample two-sided Welch tests. Gray symbols are in the absence of phage, black symbols are in the presence of phage, and yellow symbols are for internal growth. Source data are provided as a Source Data file.

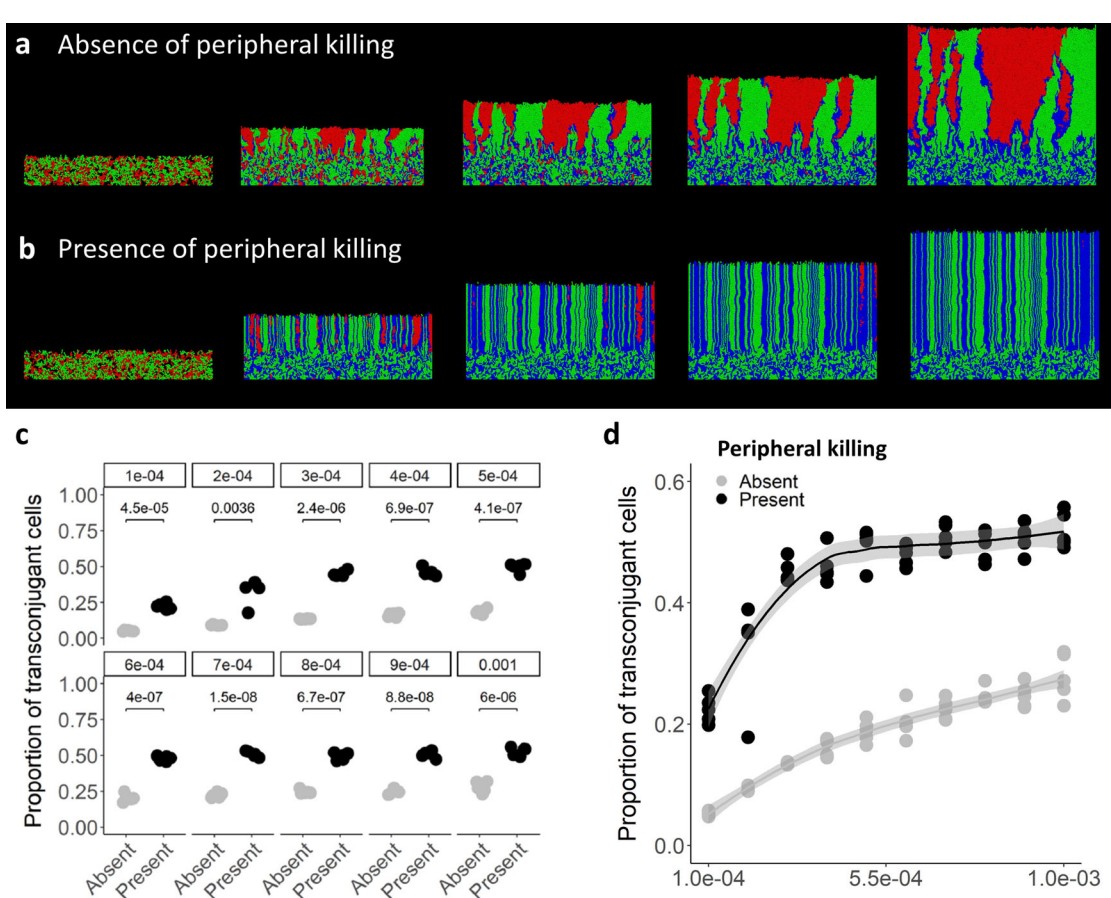

**Fig. 5 | Surface-associated growth simulations in the presence of plasmids.**
**a**, **b** Representative simulations ($n = 5$) of two competing bacterial strains where the red cells are potential recipients and the green cells carry a plasmid that reduces its growth rate by 5%. If a red cell receives the plasmid from a green cell, its growth rate is reduced accordingly and appears blue. We simulated biomass growth until reaching a population size of 18,000 cells in the (**a**) absence or (**b**) presence of peripheral killing. **c**, **d** Proportion of transconjugants within the total biomass as a function of the plasmid transfer probability. **c** Each datapoint is a measurement for an independent simulation ($n = 5$) and the $P$ values are for two-sample two-sided Welch tests. Gray symbols are in the absence of phage, black symbols are in the presence of phage. Source data are provided as a Source Data file. **d** Each datapoint is a measurement for an independent simulation ($n = 5$), the lines are the running means, and the shaded regions are the standard error. Source data are provided as a Source Data file.

limit plasmid spread[46]. Instead, we show that phage predation can increase the functional repertoires of microbial ecosystems even in the absence of positive selection for those functions. Such insights are important for understanding how functions that are deleterious for human and environmental health, such as antibiotic resistance and virulence, can persist within microbial ecosystems.

The underlying mechanism is that phage predation creates straighter interfaces between different microbial populations during surface-associated growth. This slows spatial demixing, increases cell–cell contacts, and promotes conjugation-mediated plasmid transfer. Straighter interfaces form because phage predation shifts the location of fastest microbial growth from the biomass periphery to the interior where cells have higher packing densities and are aligned closer to parallel. In contrast, cells at the biomass periphery have lower packing densities and are aligned further from parallel, which causes meandering interfaces to form that are more likely to merge together during growth and reduce spatial intermixing[28,47]. We believe this mechanism is not restricted to phage predation; rather, it should be applicable to any type of predation or inhibition that targets cells at the biomass periphery and shifts the location of fastest microbial growth to the interior.

We expect the straighter interfaces formed by phage predation to have consequences that extend well beyond plasmid transfer. Cell–cell

contacts are important for the manifestation and operation of many microbial functions and traits (e.g., metabolic cross-feeding, contact-dependent killing, etc.). For example, in marine particle-degrading communities, different bacterial populations coexist on particle surfaces and cross-feed metabolites resulting from the degradation of the particles[48]. The efficiency of such cross-feeding should improve with the closer spatial positioning of cross-feeding microorganisms, as this reduces the loss of cross-fed metabolites into the environment. A similar phenomenon occurs in soil communities that decompose organic matter[49]. We argue here that phage predation can result in closer spatial positioning of cross-feeding microorganisms and enable more efficient and prolonged ecosystem functioning, thus providing a new perspective on how phage indirectly contribute to elemental cycling.

One limitation of our study is that we only investigate a scenario where the competing strains are equally susceptible to phage predation. If one of the strains were resistant or less sensitive to phage predation, we expect the effect size to reduce. During surface-associated growth, the phage-susceptible strain will be predated on, which will cause the phage-resistant strain to increase in frequency and eventually displace the phage-susceptible strain along the biomass periphery where resources are readily available. This will eliminate intermixing between the competing strains and consequently also

eliminate plasmid transfer between them. If positive interactions and/or obligate dependencies were to occur between the strains, however, then the dynamics could be far more complex and lead to non-trivial outcomes.

We believe that our results have immediate implications for phage therapy. Phage therapy holds promise as a strategy to combat microbial infections without using chemical antibiotics, and is therefore typically assumed to not contribute to the spread of antibiotic resistance[50–52]. However, our results raise concern that the use of predatory phage can inadvertently increase the spatial intermixing of microbial populations and facilitate contact-dependent mechanisms for the horizontal transfer of antibiotic resistance determinants. As phage therapy gains traction, we need to recognize that phage predation can have unexpected consequences on microbial community dynamics, functioning, and evolution through their effects on microbial spatial self-organization.

In conclusion, our study reveals the intricate interplay between phage predation, microbial spatial self-organization, and plasmid transfer. This underscores the multifaceted nature of microbial ecosystems and necessitates a rethinking of the consequences of phage predation on microbial evolution. The pronounced effect of phage predation on plasmid transfer might be pivotal for understanding microbial adaptability in diverse environments, especially against challenges such as the spread of antibiotic resistance. Such insights are likely instrumental in shaping microbial management strategies, biotechnological endeavors, and therapeutic interventions.

## Methods

### Strains and culture conditions

We performed all experiments with *E. coli* strains TB204 and TB205[53], which are isogenic mutants derived from *E. coli* strain MG1655. Strain TB204 (MG1655 *att*P21::*PR-sfgfp*) expresses GFP while strain TB205 (MG1655 *att*P21::*PR-mcherry*) expresses mcherry from the lambda promoter (PR)[54] located on the chromosome. We introduced the self-transmissible conjugative plasmid R388 (R388 parS1-Cm), which encodes for CFP and chloramphenicol resistance[40,55,56], into strain TB204 via conjugation from *E. coli* strain DH5α using conventional filter mating on agar plates. We refer to strain TB204 carrying R388 as the R388 donor, strain TB205 as the potential recipient, and strain TB205 that receives R388 from the R388 donor as a transconjugant. We routinely cultured all strains in liquid lysogeny broth (LB) medium at 37 °C with shaking at 150 rpm. To avoid the proliferation of R388 segregants of the R388 donor, we supplemented the LB medium with 25 µg/mL chloramphenicol. For long-term preservation, we archived all strains in 15% (v/v) glycerol stocks at −80 °C. Prior to each experiment, we cultured each strain individually overnight by streaking the corresponding −80 °C stock onto an LB agar plate and used a single colony for inoculation of liquid cultures.

We used the lytic phage T6[41] for all experiments. To culture the phage, we used strain TB204 that is free of R388 as the host and incubated the culture at 37 °C for 4 h with shaking at 150 rpm. We then obtained purified phage by filtering the culture through a 0.22-µm membrane, collecting the supernatant, and storing the phage-containing supernatant at 4 °C prior to further use. For long-term storage, we mixed equal volumes of strain TB204 and phage suspensions, incubated them with shaking for 10 min at 150 rpm, and archived them in 15% (v/v) glycerol stocks at −80 °C.

### Surface-associated growth experiments

We used LB agar plates for all of our surface-associated growth experiments. We prepared agar plates by combining 25 g/L LB and 10 g/L bacteriology-grade agar powder (AppliChem, Darmstadt, Germany) with 1000 mL of distilled water and added an additional 5 mM sodium nitrate to improve growth in anoxic conditions. We then autoclaved the medium at 121 °C for 20 min, let the medium cool to 70 °C, dispensed 10 mL aliquots of the medium into sterile petri dishes with a diameter of 3.5 cm, and allowed the medium to solidify at room temperature for 2 h. Once solidified, we transferred the agar plates to a sterile hood, dried them with the lids open for 10 min, covered them with their respective lids, sealed them with Parafilm (Amcor, Zürich, Switzerland), and stored them at 4 °C until further use.

To prepare the bacterial cultures for the experiments, we first cultivated the R388 donor and potential recipient individually overnight in LB medium. We then diluted the overnight cultures by 100-fold (vol: vol) in fresh LB medium and incubated the dilution at 37 °C with shaking at 150 rpm for 4 h to ensure the cells were in the logarithmic growth phase. Thereafter, we adjusted the optical density at 600 nm ($OD_{600}$) of each culture to one in a volume of 1 mL, corresponding to ~$10^8$ colony forming units (CFU)/mL. We then washed the cells three times by centrifugation at 3600×g and 4 °C for 10 min and resuspended the washed cells in 1 mL of phosphate-buffered saline (PBS).

To prepare the phage for the experiments, we mixed the refrigerated phage stock with TB204 and incubated the mixture in a shaker at 37 °C and 150 rpm for 4 h. After incubation, we removed the TB204 cells from the solution by filtration through a 0.22-µm membrane to obtain a cell-free and active phage solution. We determined the concentration of the phage using the double-layer plate method[57] and diluted the phage solution to a concentration of $10^8$ in PBS.

For the surface-associated growth experiments, we mixed the R388 donor and potential recipient at a ratio of one (cell number:cell number) and deposited single 1 µl droplets onto the centers of separate LB agar plates. After allowing the bacteria to grow for 6 h, we added a 1 µl droplet of the phage solution or PBS (control) to the bacteria and incubated the plates in anoxic conditions for 10 days at 22 °C. To impose anoxic conditions, we incubated the plates in a glove box (Coy Laboratory Products, Grass Lake, MI, USA) containing a nitrogen:hydrogen (97:3) atmosphere. We performed ten replicates for each treatment and randomly selected five replicates that did not contain any phage-resistant mutants for subsequent imaging and quantitative analysis. Thus, we excluded the evolution of phage resistance during the time-course of the experiment, which typically occurred in one to three of the ten replicates per treatment under our experimental conditions. We provide an example of a replicate that we excluded due to the emergence of phage-resistant mutants in Supplementary Fig. 9.

### Confocal laser-scanning microscopy and image analysis

We imaged the biomass at the end of the surface-associated growth experiments using a Leica TCS SP5 II CLSM (Leica Microsystems, Wetzlar, Germany) equipped with a 5× HCX FL objective, a numerical aperture of 0.12, and a frame size of 1024 × 1024 (resulting in a pixel size of 3.027 µm). We set the laser emission to 458 nm for the excitation of CFP, to 488 nm for the excitation of GFP, and to 514 nm for the excitation of RFP. We set the emission filter to 469–489 nm for CFP, to 519–551 nm for GFP, and to 601–650 nm for RFP.

We quantified spatial intermixing using a well-established intermixing index[35,37,58,59]. We drew a circle at a specific radial distance from the inoculation droplet periphery, quantified the number of color transitions (each strain expresses a different fluorescent protein) along the circumference of the circle, and normalized the number of color transitions by the circle's circumference. We repeated this process for various radii to obtain spatial intermixing as a function of radial growth. We used Fiji (https://fiji.sc) to perform these analyses. We first used the "Mean" algorithm to threshold and binarize each image and the "remove outliers" algorithm (radius = 2, threshold = 50, bright) to eliminate noise. We next used the Sholl plugin[60] of ImageJ to apply a concentric windowing method from the inoculation droplet periphery to the final biomass periphery at a radial increment of 5 µm. We then calculated the intermixing index ($I_r$) for each concentric window as the

 

number of color transitions ($N_r$) divided by the expected number of color transitions for a random spatial distribution of two strains at a specific radius ($r$), using Eq. (1).

$$I_r = \frac{N_r}{\pi r/2} \qquad (1)$$

## Scanning electron microscopy

To perform SEM, we first exposed the biomass on the agar plates to 5% glutaraldehyde vapors and then to 2% osmium tetroxide vapors. After fixation, we cut the biomass out of the agar plate and immersed it in 30% ethanol followed by further incubations in 50%, 70%, 90%, and 100% ethanol. After several incubations in absolute-grade ethanol, we placed the biomass samples in metal capsules and exchanged the ethanol with liquid $CO_2$ using a CPD 931 critical point dryer (Tousimis Research, Rockville, MD, USA). We next raised the temperature and pressure above the critical point of $CO_2$, released the pressure, and mounted the biomass samples onto SEM stubs with conductive carbon cement. Finally, we sputter coated the biomass samples with 4 nm Pt/Pd with planetary rotation using a CCU-010 vacuum coating system (Safematic, Zizers, Switzerland) and imaged the samples with a TFS Magellan 400 SEM (Thermos Fisher Scientific, Waltham, United States) at 2 kV and secondary electron detection.

## Individual-based computational modeling

We used the open-source modelling software Cellmodeller 4.3 framework (https://github.com/cellmodeller/CellModeller) and the "MicrobialEcologyToolbox" branch to perform individual-based simulations of surface-associated microbial growth[61,62]. We wrote sections of the code with the aid of ChatGPT-4 (https://openai.com/gpt-4). In our simulations, we represent bacterial cells as three-dimensional capsules of length $L$ that grow uniaxially. After reaching a specified target length, each cell divides into two daughter cells that inherit the properties of their parent. We set the biophysical parameter "gamma", which adjusts the ratio between the drag force on cell translation relative to cell growth, to 20. This parameter affects the rate at which cells stop growing due to constraints from physical forces. During each time step, each cell expends energy to grow and displace neighboring cells. If surrounding cells significantly impedes this process, the cell stops growing.

To simulate *E. coli* cells, we set the initial cell shape to a radius of 0.5 μm and a length of 2 μm. Each cell divides into two when its length reaches a random value between 3.5 and 4.0 μm. We set the cell growth rate to 1, which means that each cell would ideally grow 1 micron per unit time step in the absence of physical constraints. We began the simulations with 600 cells of each type (1200 cells in total). We restricted the growth region such that cells only grow in the space from −100 units to within 100 units along the x-axis and *y* axis starting from 0 units upwards, which means that cells only grow in the positive direction along the *y* axis once they have filled the initial space. We assigned the initial position of each cell randomly within the specified 2D space with its x-coordinate in the range [−100, 100] and y-coordinate in the range [0, 30]. We assigned the initial rotation of each cell randomly with the *x* and *y* components of the direction vector within the range [−1, 1].

To simulate conjugation-mediated plasmid transfer, we used the *CompNeighbours* function in CellModeller to detect cell contacts and calculate the probability of successful plasmid transfer after contact using a constant probability of 0.001 to 0.01 per simulation timestep. If successful, the potential recipient cell becomes a transconjugant cell and changes color to blue. The new transconjugant cell can then conjugate with a new potential recipient cell. We set the specific growth rate of plasmid donor and transconjugant cells to 0.95 to account for the cost of carrying the plasmid[42]. As we did not use

antibiotics or apply any positive selection for the plasmid, there was never an advantage to carrying the plasmid.

To investigate the phenomenological effect of phage predation on surface-associated growth and plasmid transfer, we used the *killflag* feature in CellModeller. This feature allows the removal of cells from the model and subsequently updates the list of cell states, keeping only live cells. Rather than directly simulating the complex biophysical process of phage replication and predation, we represent the phage as a signal that initiates cell removal. Hence, we do not explicitly simulate phage particles; rather, we simulate the expected phage-induced killing of cells located at the biomass periphery, which is based on prior work that simulates individual phage particles[31]. During the simulation as the biomass grows along the *y* axis, we extracted the *y* axis values of all cell locations. We then identified the cell with the largest *y* axis value as the peripheral cell and define the phage-infection region as those cells within 4 μm of the peripheral cell. Finally, we eliminated cells within this region at a specific frequency to simulate the effect of phage predation.

We performed all simulations using a 2021 Windows system ThinkPad laptop computer with concurrent simulations distributed between a Platform Intel(R) 2.80 GHz Core(TM) i7-1165G7 OpenCL HD Graphics and device Intel(R) Iris(R) Xe Graphics. We stored the status and spatial location of each cell every 20 time-steps and we performed simulations until reaching 18,000 cells. We launched the Graphical User Interface (GUI) by running a written script and saved the Pickle data from the GUI. We performed each simulation five independent times. All parameters are listed in Supplementary Table 1.

## Intermixing index of computational modeling simulations

We calculated the intermixing index in the simulations (denoted as $I_{simulation}$) using Eq. (2).

$$I_{simulation} = \frac{1}{N} \sum_{i=0}^{N} \frac{\sum_{j=0}^{n(i)} I(i,j)}{n(i)} \qquad (2)$$

In Eq. (2), $N$ is the total number of cells and $n(i)$ is the number of neighboring cells i. Neighboring cell I(i,j) yields a value of either 1 or −1, contingent upon the neighboring cell possessing a distinct (1) or identical (−1) cell type compared to the focal cell. For an individual cell i, we iterated across all neighboring cells that are in physical contact j using Eqs. (3) and (4).

$$I = 1 \text{ if } cellType[i] \neq cellType[j] \qquad (3)$$

$$I = -1 \text{ if } cellType[i] = cellType[j] \qquad (4)$$

This methodology aligns with the spatial assortment parameter and segregation index[63]. Regions dominated by isogenic cell clusters, where the majority of cells are adjacent to those of the same type, have a negative intermixing index. Conversely, regions with frequent intermixing between distinct types have a positive intermixing index ($I_{simulation}$).

## Quantification of interface straightness

We calculated the interface straightness for both the experiments and simulations using Python 3.11. We first used the Canny edge detection method[64] to identify the interfaces using a lower threshold of 100 and an upper threshold of 200. To focus the analysis on specific regions of the images, we defined an annular (ring-shaped) region on each image, where we positioned the center of the annulus at the geometric center of each image. We then extracted the interfaces within the annular region using the findContours function from the OpenCV library and only considered those that touched both the interior and peripheral boundaries of the annular region. We used a tolerance of five pixels to

account for minor variations. We then quantified the straightness of the interfaces as follows: For each interface, we drew a straight line connecting the beginning and end points. We then calculated the average perpendicular distance of all points on the interface to the straight line connecting the beginning and end points using Eq. (5).

$$S = \frac{d_{max}}{d_{max} + d_{avg}} \tag{5}$$

In Eq. (5), S is the straightness, $d_{max}$ is the distance between the beginning and end points on the line, and $d_{avg}$ is the average perpendicular distance of all points on the interface to the straight line connecting the beginning and end points. This returns a value between 0 and 1, where a value closer to 1 indicates that the interface is straighter.

## Statistics and reproducibility

We performed all statistics in R (v4.1.2) (https://cran.r-project.org). We used two-sample two-sided Welch tests to test for differences between means. We therefore did not make any assumptions regarding the homoscedasticity of our datasets. We used the Holm–Bonferroni method to adjust $P$ values for multiple comparisons. We used the Wilk–Shapiro test to test whether our datasets deviate from normality with a significance level of $P > 0.05$. We did not observe any deviations from normality for any of our datasets. We reported the statistical test, $P$ value, and sample size ($n$) for each test in "Results".

## Reporting summary

Further information on research design is available in the Nature Portfolio Reporting Summary linked to this article.

## Data availability

All raw numerical and image data generated in this study have been deposited in the Eawag Research Data Institutional Collection (ERIC) and are freely available to the public at https://doi.org/10.25678/000C1F. Source data are provided with this paper.

## Code availability

All code generated in this study have been deposited in the Eawag Research Data Institutional Collection (ERIC) and are freely available to the public at https://doi.org/10.25678/000C1F.

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

## Acknowledgements

The authors thank Dr. Maria Pilar Garcillán-Barcia, Instituto de Biomedicina y Biotecnología, University of Cantabria, for providing plasmid R388; Dr. Martin Ackermann, Swiss Federal Institute of Aquatic Science and Technology, for providing *E. coli* strains TB204 and TB205; and Zachary Bailey, Department of Environmental Systems Science, ETH Zürich, for providing phage T6. The authors thank Dr. Anne Greet Bittermann, Scientific Center for Optical and Electron Microscopy, ETH Zürich, for assistance with electron microscopy. We thank Mrs. Ruan (Guo Chen) for her excellent assistance with drawing the schematic diagram in Fig. 1. This work was supported by a grant of the Swiss National Science Foundation (310030_207471) to D.R.J., by grants of the National Key Research and Development Project of China (2022YFD15002005), the National Natural Science Foundation of China (42277298), the 2115 Talent Development Program of the China Agricultural University (1191-00109012), and the National High-end Foreign Experts Recruitment Plan (G2022108011L) to G.W., and by a grant of the Swiss National Science Foundation (P2EZP3_199849) to J.R.

## Author contributions

C.R., G.W., and D.R.J. conceived the study. C.R. and D.R.J. designed the experiments. C.R. performed the experiments. C.R., A.K., and T.R.J. developed the individual-based computational model. C.R. performed all of the simulations. C.R. and J.R. analyzed the data. C.R. and D.R.J. wrote the manuscript with feedback from J.R., G.W., and A.K.

## Competing interests

The authors declare no competing interests.
