## [Peer Review File · Nature Communications]

REVIEWER COMMENTS

Reviewer #1 (Remarks to the Author):

This is a very good paper with a clear hypothesis relating phage predation to the slowdown of demixing of bacteria and therefore to the increase of plasmid spread, with compelling microscopy demonstrating the effect.

My major issue is with the modeling section – in which the authors describe their use of an Individual-Based Modeling approach (via CellModeler) to describe how phage impact their cell population dynamics. The issue is that there is no real way to include individual phage without significant slowdown of the entire agent based model. Hence, no phage were included in the model. Instead a ‘killflag’ feature was used as a proxy. But, how do the authors know that the phage behave like a killflag only killing off peripheral cells within 4 microns of the peripheral cell? Moreover, why are a fixed frequency removed? And, for real phage, there would be amplification of the phage signal. In essence, the beautiful experiments in Figures 2-3 are augmented by simulations in Figures 4-5 that make highly non-realistic assumptions without justification that nonetheless reproduce elements seen in experiments. On the plus side, I think the simulations could suggest that any kind of killing mechanism near the periphery could slowdown the demixing effect and then increase plasmid transfer – this is a major point of the authors. But, on the negative side, the Results section makes it seem as if the Model actually includes an explicit representation of phage (see lines 211-213): “We formulated our model such that phage predate on cells located at the biomass periphery and observe the same effect as in our experiments. When phage are absent, cells at the biomass periphery grow fastest...”

This is factually incorrect. There are no phage in the model. There is a killflag – there is no replication/release of new phage/diffusion of phage/etc. The paper’s Abstract/Results/Discussion should address literally what the authors did on the simulation side, which is a ‘generic’ peripheral-based killing that is meant to represent phenomenologically the hypothesized impact of phage killing without explicitly representing phage dynamics. I think it may be a good choice, but the authors should be more forthcoming about what they did and then go over caveats/limitations/future work to make such representations explicit in the Discussion.

Minor comments

Do not accept the null hypothesis. There are a few statements with $p > 0.05$ in which the authors accept the null of no effect rather than saying that there was not statistical evidence of an effect, e.g., “We find that DNase I has no effect on the number of transconjugants...” (lines 146-7)

Reviewer #2 (Remarks to the Author):

The authors investigate the impact of phage predation on plasmid conjugation in spatially structured environments and show that, surprisingly, phage predation increases the number of transconjugants despite reducing the overall number of bacteria. They go on to show that this facilitation stems from phage predation slowing the demixing of strains, which usually occurs in surface-associated spread. Through a combination of experiments and modelling they show that a shift of fastest growth to the biomass interior (as for example caused by phage predation on the periphery) leads to straighter interfaces between the donor and recipient strain and more interaction surfaces for conjugation.

This study is novel and comprehensive, exploring the interaction mechanisms of two types of mobile genetic elements and their unintuitive effects on bacterial evolution. The paper is also well written and I only have minor comments to add.

- The text is well written and easy to follow, but the abstract could be made more accessible as it uses language that is hard to follow without the explanations given later on, e.g. rotationally ordered, coalesce, spatial de-mixing. The link between L35-37 and the next sentence is for example unclear as it is based on understanding what rotationally ordered means and how that will affect the geometry of the interfaces. Also, is 'the use of predatory phage to control microbial proliferation' in the first sentence referring to phage therapy? If so, it might be easier to use that term. If it is meant to be more generally about phage predation on bacteria, I would replace the word 'control'.

- I am missing a bit more discussion on what we already know about how phages affect plasmid spread: e.g. Harrison et al., 2015 (mBio); Iglar et al., 2022 (Phil.Trans.B); Olaja et al., 2013 (Evol.Appl.)

- It would be great to have a paragraph of discussion on the fact, that the effects seem to assume equal susceptibility of both strains to the phage and how the authors expect the effect to change if the strains have different phage susceptibility, which is likely to occur in natural microbial communities.

- Similarly, and this is more curiosity, would you expect the effect to be dependent on phage parameters like adsorption rate and burst size, i.e. would you expect a gradual shift from random patterns to straight interfaces or would that be more of a threshold effect as soon as phage virulence is high enough?

- L237: This title feels quite repetitive from what was already show in L106-L130. Maybe focus more on the new results in that section, e.g. that phage presence lowers the necessary magnitude of the conjugation rate to make up for plasmid costs.

- The Figure 1 legend should probably say T6 not T7.

Reviewer #2 (Remarks on code availability):

The link did not work

Reviewer #3 (Remarks to the Author):

In this manuscript, Ruan and co-authors show that phage predation shapes bacterial spatial distribution, which facilitates the spread of conjugative plasmids. This is an interesting work, both from a eco-evolutionary perspective but also important for applications such as phage therapy. I found it scientifically sound, as well as elegantly presented.

I am generally convinced by the authors work and claims. Despite that, I have one main comment. The first results sections show how phage predation affects the spread of R338 as measured as number of transconjugant areas and as a proportion of biomass. However, it is also shown that phage predation reduces total biomass. Therefore, I wonder how the latter effect was corrected for. Because with lower total biomass the area/amount of transconjugants would be relatively larger, since the regions that tend to coalesce the most are the ones more distant from the centre of the population. Shouldn't then the authors compare only identical areas (same radius size) to avoid artefacts?

I don't think this will change the conclusions, but should be a control the provides more robustness to the work.

Another point is related to the measurement of intermixing and demixing. I see that the quantification is explained in the methods section, but I think it would make it easier to the reader if a brief explanation was also presented in the main text (e.g., lines 158-160). I admit that I may not have understood the method entirely.

And a few minor comments:

- line 136: "E. coli" should be in italics.

- line 347: "the lambda promotor", since the phage encodes several genes/promotors, which promotor are the authors referring to?

- line 348/9: "R388 [...] encodes chloramphenicol resistance [40]". The original plasmid is resistant to trimethoprim (not chloramphenicol) so I presume the authors are working with a derivative. I looked through reference 40 but it seems to concern plasmid R1. Could the authors provide a reference for the specific R388 they used?

14th May, 2024

We would like to thank you and all three reviewers for the time and effort put into evaluating and improving the quality and clarity of our manuscript. We are pleased with the positive assessment of our work and the thoughtful and constructive suggestions for its improvement. We have now revised the manuscript accordingly and have addressed all of the points that were raised. We believe that our manuscript has greatly improved from the revision process and we hope that it is now suitable for publication in *Nature Communications*.

All line numbers refer to those in the revised version of the main text. Reviewer comments are shown in normal font. Author responses are in blue font. Specific changes to the manuscript are indicated in blue italic font.

Sincerely and on behalf of all the co-authors,

Dave Johnson

Reviewer #1 (Remarks to the Author):

This is a very good paper with a clear hypothesis relating phage predation to the slowdown of demixing of bacteria and therefore to the increase of plasmid spread, with compelling microscopy demonstrating the effect.

We thank the reviewer for their positive assessment of our manuscript and we are pleased that the reviewer found our study to be clear and compelling.

My major issue is with the modeling section – in which the authors describe their use of an Individual-Based Modeling approach (via CellModeler) to describe how phage impact their cell population dynamics. The issue is that there is no real way to include individual phage without significant slowdown of the entire agent based model. Hence, no phage were included in the model. Instead a ‘killflag’ feature was used as a proxy. But, how do the authors know that the phage behave like a killflag only killing off peripheral cells within 4 microns of the peripheral cell? Moreover, why are a fixed frequency removed? And, for real phage, there would be amplification of the phage signal. In essence, the beautiful experiments in Figures 2-3 are augmented by simulations in Figures 4-5 that make highly non-realistic assumptions without justification that nonetheless reproduce elements seen in experiments. On the plus side, I think the simulations could suggest that any kind of killing mechanism near the periphery could slowdown the demixing effect and then increase plasmid transfer – this is a major point of the authors. But, on the negative side, the Results section makes it seem as if the Model actually includes an explicit representation of phage (see lines 211-213): “We formulated our model such that phage predate on cells located at the biomass periphery and observe the same effect as in our experiments. When phage are absent, cells at the biomass periphery grow fastest...” This is factually incorrect. There are no phage in the model. There is a killflag – there is no replication/release of new phage/diffusion of phage/etc. The paper’s Abstract/Results/Discussion should address literally what the authors did on the simulation side, which is a ‘generic’ peripheral-based killing that is meant to represent phenomenologically the hypothesized impact of phage killing without explicitly representing phage dynamics. I think it may be a good choice, but the authors should be more forthcoming about what they did and then go over caveats/limitations/future work to make such representations explicit in the Discussion.

We greatly appreciate this comment and for highlighting that our presentation was misleading. Indeed, we do not model phage explicitly. Rather, we apply a kill flag that is meant to capture the phenomenological effect of phage predation on a growing bacterial colony. Note that our use of this kill flag is based on prior work by Erikson and co-workers (<https://www.pnas.org/doi/full/10.1073/pnas.1708954115>). In that work, the authors explicitly simulated individual phage particles and found that phage predation results in peripheral killing, where the depth of penetration of killing is related to population sizes and phage properties. Thus, rather than simulate individual phage particles, which is extremely computationally expensive and would limit the number of bacterial cells that we could simulate, we assume that phage predation results in peripheral killing as demonstrated by Erikson and co-workers. We now state this in the following lines of the revised manuscript.

Lines 217-222: *“To test this, we modified and employed an individual-based computational model where we assume that phage predation results in peripheral killing of the bacterial biomass. We implemented this approach because a prior study that simulated individual phage particles found that phage predation results in peripheral killing, where the depth of killing is related to population sizes and phage properties³¹. Using this approach, we find that peripheral killing accurately reproduces the effects that we observe in our experiments.”*

To further clarify this, we changed the term “phage” to “peripheral killing” throughout the entirety of the manuscript whenever discussing the model simulations. This includes the main text, figures, figure legends, extended figures, and movies. For example, see these following lines of the revised manuscript (note that this is only an example and we have applied this change throughout).

Lines 222-223: *“When peripheral killing is absent in our simulations, cells at the biomass periphery grow the fastest.”*

We also restated our approach in the introduction to make it clear that we are not explicitly simulating phage predation in the following lines of the revised manuscript.

Lines 102-105: *“We complement our experiments with individual-based computational simulations that test how the peripheral killing caused by phage predation can reshape microbial spatial organization and increase plasmid transfer during surface-associated growth.”*

Finally, we now make it clear in the methods that we do not explicitly simulate phage particles in the following lines of the revised manuscript.

Lines 512-514: *“Hence, we do not explicitly simulate phage particles; rather, we simulate the expected phage-induced killing of cells located at the biomass periphery, which is based on prior work that did simulate individual phage particles.”³¹*

With regards to the limitations of our simulation approach, please see our response to Reviewer 2 who had a similar curiosity. Briefly, our approach does not allow us to explicitly simulate phage properties, such as the adsorption rate or burst size. However, we do not expect these properties to affect the main qualitative outcome. The adsorption rate and burst size will affect the penetration depth of killing along the biomass periphery. However, this will not change the fact that phage preferentially predate on cells located along the biomass periphery (governed by fundamental principles of mass transfer), which will shift the point of maximum growth to the interior and lead to straighter interfaces and increased plasmid transfer. Thus, we do not expect that simulating individual phage particles will change our main conclusions.

Minor comments

Do not accept the null hypothesis. There are a few statements with $p > 0.05$ in which the authors accept the null of no effect rather than saying that there was not statistical evidence of an effect, e.g., “We find that DNase I has no effect on the number of transconjugants...” (lines 146-7)

Changed as recommended. We have revised these statements such that we no longer accept the null hypothesis in the following lines of the revised manuscript.

Lines 149-152: *“We do not find statistically significant evidence that DNase I affects the number of transconjugants (two-sample two-sided Welch test; $P = 0.38$, $n = 5$) (Extended Data Fig. 3b), which is expected if natural transformation is negligible.”*

Reviewer #2 (Remarks to the Author):

The authors investigate the impact of phage predation on plasmid conjugation in spatially structured environments and show that, surprisingly, phage predation increases the number of transconjugants despite reducing the overall number of bacteria. They go on to show that this facilitation stems from phage predation slowing the demixing of strains, which usually occurs in surface-associated spread. Through a combination of experiments and modelling they show that a shift of fastest growth to the biomass interior (as for example caused by phage predation on the periphery) leads to straighter interfaces between the donor and recipient strain and more interaction surfaces for conjugation. This study is novel and comprehensive, exploring the interaction mechanisms of two types of mobile genetic elements and their unintuitive effects on bacterial evolution. The paper is also well written and I only have minor comments to add.

We thank the reviewer for their positive assessment of our manuscript and we are pleased that the reviewer found our study to be novel, comprehensive, clear, and significant.

- The text is well written and easy to follow, but the abstract could be made more accessible as it uses language that is hard to follow without the explanations given later on, e.g. rotationally ordered, coalesce, spatial de-mixing. The link between L35-37 and the next sentence is for example unclear as it is based on understanding what rotationally ordered means and how that will affect the geometry of the interfaces. Also, is 'the use of predatory phage to control microbial proliferation' in the first sentence referring to phage therapy? If so, it might be easier to use that term. If it is meant to be more generally about phage predation on bacteria, I would replace the word 'control'.

We thank the reviewer for these suggestions to improve the clarity and flow of the text and to make the abstract more accessible to a broader audience. We have taken the following actions:

- 1) We changed the term "*rotationally ordered*" to "*aligned closer to parallel*" throughout the entirety of the manuscript.
- 2) We changed the term "*coalesce*" to "*merge together*" throughout the entirety of the manuscript.
- 3) We changed the term "*spatial demixing*" to "*spatial segregation*" in the abstract and defined spatial demixing more precisely in the main text.
- 4) Regarding the unclear linkage in L35-37 due to the term "*rotationally ordered*", we amended these lines as follows.

Lines 39-41: *This creates straighter interfaces between the strains that are less likely to merge together during growth, consequently slowing the spatial segregation of the strains and enhancing plasmid transfer between them.*

- 5) We also reconsidered the phrasing "the use of predatory phage to control microbial proliferation" in the abstract and agree that it could be misconstrued. Our intent was that the term phage predation would encapsulate both the natural ecological process and engineered applications (e.g. phage therapy). However, by starting the sentence with "The use of...", we gave the impression that we were only talking about applications. We have now rephrased this to ensure that we encapsulate both natural processes and applications.

Lines 29-30: *"Phage predation is generally assumed to reduce microbial proliferation while not contributing to the spread of antibiotic resistance."*

- I am missing a bit more discussion on what we already know about how phages affect plasmid spread: e.g. Harrison et al., 2015 (mBio); Iglar et al., 2022 (Phil.Trans.B); Olaja et al., 2013 (Evol.Appl.)

We thank the reviewer for pointing out additional relevant literature. We have now expanded the discussion of existing knowledge regarding how phage predation can slow plasmid spread, with particular reference to the studies of Harrison et al. (2015), Iglar et al. (2022) and Olaja et al. (2013).

Lines 289-292: *"For example, phage predation can slow plasmid spread by creating an additional death rate⁴⁴, by specifically targeting cells that express plasmid-encoded traits⁴⁵, and by modifying selection pressures that limit plasmid spread⁴⁶."*

- It would be great to have a paragraph of discussion on the fact, that the effects seem to assume equal susceptibility of both strains to the phage and how the authors expect the effect to change if the strains have different phage susceptibility, which is likely to occur in natural microbial communities.

We agree that an important next step is to consider systems where the component strains do not have equivalent susceptibility to phage predation. Our expectation is that if one strain is less sensitive or resistant to the phage, then

the effect size will reduce. Consider two competing strains where one is phage-susceptible while the other is phage-resistant. When propagating across a surface, the phage-susceptible strain will be predated on, which will allow the phage-resistant strain to increase in frequency and eventually displace the phage-susceptible strain along the biomass periphery. This will eliminate intermixing between the strains and consequently also eliminate plasmid transfer between them. Of course, if positive interactions and/or obligate dependencies were to occur between the strains, then the dynamics would be far more complex and likely lead to non-trivial outcomes. We now discuss this in the following lines of the revised manuscript.

Lines 323-331: *“One limitation of our study is that we only investigate a scenario where the competing strains are equally susceptible to phage predation. If one of the strains were resistant or less sensitive to phage predation, we expect the effect size to reduce. During surface-associated growth, the phage-susceptible strain will be predated on, which will cause the phage-resistant strain to increase in frequency and eventually displace the phage-susceptible strain along the biomass periphery where resources are readily available. This will eliminate intermixing between the strains and consequently also eliminate plasmid transfer between them. If positive interactions and/or obligate dependencies were to occur between the strains, however, then the dynamics could be far more complex and lead to non-trivial outcomes.”*

- Similarly, and this is more curiosity, would you expect the effect to be dependent on phage parameters like adsorption rate and burst size, i.e. would you expect a gradual shift from random patterns to straight interfaces or would that be more of a threshold effect as soon as phage virulence is high enough?

This is indeed an interesting curiosity that we have thought about in some detail. We expect that the adsorption rate and burst size will have quantitative effects on spatial organization and plasmid transfer but will not change the main qualitative outcome. Briefly, as the adsorption rate and the burst size increase, the depth of killing along the biomass periphery will also increase. This will shift the location of maximum growth deeper into the biomass where local cell densities are even higher, which will cause the formation of even straighter interfaces. However, the adsorption rate and burst size will not change the fact that phage preferentially predate on cells located along the biomass periphery (governed by fundamental principles of mass transfer), and they should therefore not change the general expectation that phage predation causes the formation of straighter interfaces and increases plasmid transfer.

- L237: This title feels quite repetitive from what was already show in L106-L130. Maybe focus more on the new results in that section, e.g. that phage presence lowers the necessary magnitude of the conjugation rate to make up for plasmid costs.

Changed as recommended to the following.

Lines 249: *“Peripheral killing lowers the conjugation rate needed to compensate for plasmid cost”*

- The Figure 1 legend should probably say T6 not T7.

Changed to T6.

Remarks on code availability: The link did not work

We apologize that the link was not working (it was correct in the reporting document but not in the main text). We have now updated the link in the main text. The link is <https://drive.switch.ch/index.php/s/ozZJPqLLrvVLbBr>. We also reiterate that we will make all data and code publicly available in the ERIC institutional repository of Eawag upon acceptance.

Reviewer #3 (Remarks to the Author):

In this manuscript, Ruan and co-authors show that phage predation shapes bacterial spatial distribution, which facilitates the spread of conjugative plasmids. This is an interesting work, both from an eco-evolutionary perspective but also important for applications such as phage therapy. I found it scientifically sound, as well as elegantly presented.

We thank the reviewer for their positive assessment of our manuscript and we are pleased that the reviewer found our study to be interesting, scientifically sound, and elegantly presented.

I am generally convinced by the authors work and claims. Despite that, I have one main comment. The first results sections show how phage predation affects the spread of R338 as measured as number of transconjugant areas and as a proportion of biomass. However, it is also shown that phage predation reduces total biomass. Therefore, I wonder how the latter effect was corrected for. Because with lower total biomass the area/amount of transconjugants would be relatively larger, since the regions that tend to coalesce the most are the ones more distant from the centre of the population. Shouldn't then the authors compare only identical areas (same radius size) to avoid artefacts? I don't think this will change the conclusions, but should be a control that provides more robustness to the work.

We agree with the reviewer that this would add robustness to the study and main conclusions. We now include the results comparing the total number of transconjugants measured at a fixed radial distance of 2100 μm , which is the minimum radius that is present across all of the phage-treated and untreated samples. This data is presented in a new Extended Data Figure (Extended Data Fig. 1). When using this quantity, we again find that the number of transconjugants is significantly higher in the phage-treated samples, which adds robustness to our main conclusions.

Extended Data Fig. 1: Total transconjugant area divided by the total biomass area at a fixed radial distance from the biomass centroid. We calculated the number of transconjugants using the same images that we used for Fig. 2 in the main text. We selected a fixed radial distance of 2100 μm , which is the minimum distance that is present across all of our samples. Each datapoint is a measurement for one of five independent experimental replicates and the P value is for a two-sample two-sided Welch test.

We now refer to this new Extended Data Figure in the following lines of the revised manuscript.

Lines 121-123: “Finally, our outcomes remain valid when we calculate the number of transconjugants at a fixed radial distance across all of our samples (Extended Data Fig. 1).”

Another point is related to the measurement of intermixing and demixing. I see that the quantification is explained in the methods section, but I think it would make it easier to the reader if a brief explanation was also presented in the main text (e.g., lines 158-160). I admit that I may not have understood the method entirely.

Added as recommended. This description is in the following lines of the revised manuscript.

Lines 163-168: “We use an intermixing index that quantifies the number of transitions between two strains (colors) adjusted for the anticipated number of transitions for a random spatial arrangement of the two strains. We first position a circle with its center located at the centroid of the biomass. We then calculate the intermixing index along the circumference of the circle as $I_r = (2N_r)/\pi r$, where r is the radius of the circle. This allows us to quantify the intermixing index as a function of the radial extent of growth.”

And a few minor comments:

- line 136: "E. coli" should be in italics.

Changed as recommended.

- line 347: "the lambda promotor", since the phage encodes several genes/promoters, which promotor are the authors referring to?

Clarified as recommended. The lambda promotor in the strain TB204 and TB205 is lambda promoter (*PR*) according to the paper: <https://doi.org/10.1016/j.jbiotec.2018.01.008>. We have added this information in the following lines of the revised manuscript.

Lines 372-374: “Strain TB204 (MG1655 *attP21::PR-sfgfp*) expresses GFP while strain TB205 (MG1655 *attP21::PR-mcherry*) expresses mcherry from the lambda promoter (*PR*)⁵⁴ located on the chromosome.

- line 348/9: "R388 [...] encodes chloramphenicol resistance [40]". The original plasmid is resistant to trimethoprim (not chloramphenicol) so I presume the authors are working with a derivative. I looked through reference 40 but it seems to concern plasmid R1. Could the authors provide a reference for the specific R388 they used?

We apologies for the lack of clarity about plasmid R388. The chloramphenicol resistant plasmid R388 (R388 *parS1-Cm*) we used is a derivative of plasmid *par145* (pSU2007 *aph::cat-PA1/04/03-cfp*-T0*), whose construction was reported in the following study: <https://www.sciencedirect.com/science/article/pii/S0147619X12000054>. However, the chloramphenicol resistance was added to the derivative in another study from the laboratory at the University of Cantabria that shared the plasmid with us, published here (see Table S1): <https://journals.plos.org/plosgenetics/article?id=10.1371/journal.pgen.1002073#pgen.1002073.s005>. We have made this clarification in the following lines of the revised manuscript.

Lines 375-377: “We introduced the self-transmissible conjugative plasmid R388 (R388 *parS1-Cm*), which encodes for CFP and chloramphenicol resistance^{40,55,56}, into strain TB204 via conjugation from *E. coli* strain DH5 α using conventional filter mating on agar plates.”

REVIEWERS' COMMENTS

Reviewer #1 (Remarks to the Author):

Thank you for the thoughtful and clarifying changes. I appreciate the fact that the model presentation now explicitly addresses what the authors did and did not do. The experimental study is elegant - and the computational results support them. As the authors point out, more work will be needed to understand the impacts of phage feedback when treated explicitly. I liked the work in the first submission and like it even more now.

Reviewer #2 (Remarks to the Author):

The authors have addressed all reviewer comments appropriately and exhaustively. The manuscript presents an exciting contribution to the field of MGE research.

Reviewer #2 (Remarks on code availability):

The resource file contains a README file with instructions for installations and running the code.